# Detection of Potentially Compromised Computer Nodes and Clusters Connected on a Smart Grid, Using Power Consumption Data

**DOI:** 10.3390/s20185075

**Published:** 2020-09-07

**Authors:** Mohammed Almshari, Georgios Tsaramirsis, Adil Omar Khadidos, Seyed Mohammed Buhari, Fazal Qudus Khan, Alaa Omar Khadidos

**Affiliations:** 1Department of Information Technology, Faculty of Computing and Information Technology, King Abdulaziz University, Jeddah 21589, Saudi Arabia; akhadidos@kau.edu.sa (A.O.K.); mesbukary@kau.edu.sa (S.M.B.); fqkhan@kau.edu.sa (F.Q.K.); 2 Department of Information Systems, Faculty of Computing and Information Technology, King Abdulaziz University, Jeddah 21589, Saudi Arabia; aokhadidos@kau.edu.sa

**Keywords:** smart grid, power consumption, IoT, machine learning, security, privacy, descriptive analysis, F-Test of two samples of variance, two-way analysis of variance (ANOVA), autoregressive integrated moving average (ARIMA), malware detection

## Abstract

Monitoring what application or type of applications running on a computer or a cluster without violating the privacy of the users can be challenging, especially when we may not have operator access to these devices, or specialized software. Smart grids and Internet of things (IoT) devices can provide power consumption data of connected individual devices or groups. This research will attempt to provide insides on what applications are running based on the power consumption of the machines and clusters. It is therefore assumed that there is a correlation between electric power and what software application is running. Additionally, it is believed that it is possible to create power consumption profiles for various software applications and even normal and abnormal behavior (e.g., a virus). In order to achieve this, an experiment was organized for the purpose of collecting 48 h of continuous real power consumption data from two PCs that were part of a university computer lab. That included collecting data with a one-second sample period, during class as well as idle time from each machine and their cluster. During the second half of the recording period, one of the machines was infected with a custom-made virus, allowing comparison between power consumption data before and after infection. The data were analyzed using different approaches: descriptive analysis, F-Test of two samples of variance, two-way analysis of variance (ANOVA) and autoregressive integrated moving average (ARIMA). The results show that it is possible to detect what type of application is running and if an individual machine or its cluster are infected. Additionally, we can conclude if the lab is used or not, making this research an ideal management tool for administrators.

## 1. Introduction

Unwanted applications cause many hurdles on easily accessible computers in the lab. Some common hurdles, such as slowness of the computers and security risks are caused due to unwanted background processing. Sometimes the machines maybe used remotely or at certain times (e.g., while a lab is closed) run applications that are not supposed to. It is common for users to use the computational power of company machines for personal benefits at the expense of their organization (e.g., cryptocurrency mining, hosting applications, etc.). In some cases, privacy rules may make it difficult to conduct active monitoring of what applications the users are running. Furthermore, users may use their personal machines in order to avoid detection since the administrators will have no access to these machines. However, organizations have full access to their smart power grid and they have every right to monitor the power consumption for the various computer/clusters or even rooms [1]. Smart grids have a number of advantages such as unit/cluster power consumption monitoring, automatic control by including hardware and software monitoring and controlled applications [2]. Additionally, it is possible to monitor the power consumption in real time [3].

This research will attempt to detect if a machine and a cluster of machines is infected by unwanted applications, such as a virus. It will also attempt to predict what applications/types of applications are running on the machine based on power consumption data. Finally, we can detect possible occupancy with is traditionally done by use of multiple sensors and applications [4]. The novelty of this work can be summarized by the following objectives:(1)Detect if a node and/or a cluster is infected;(2)Differentiate between different types of applications;(3)Detect occupancy of a node and/or a cluster.

This is not the first attempt to use power consumption data in order to produce diagnostics. Several researchers have focused on the use of power consumption data and used machine learning algorithms to analyze them. Some of the applied research in this field has focused on various applications in different domains such as smartphones, computers and home electricity.

Luckett et al. [5] proposed a new method for detecting any root-kit behavioral process. They were able to achieve high accuracy and high training speed due to the small size of the data set. Their data was collected from the computer processor power consumption with sample period of five minutes, then various machine learning algorithm were used and tested the method on more than one operating system to ensure that it will have high accuracy. An experiment conducted by Jimenez et al. [6] focused on collecting the power consumption data of the computer processing unit with the network data. They then used ten machine learning techniques to analyze and classify if a computer was infected or not. They used a software to generate high utilization to add more features to avoid wrong alarms. They successfully detected malwares using both network and power data using the Random Forest algorithm. Another experiment proposed by Loanes et al. [7] aimed to evaluate and detect the abnormal state of the power grid by measuring the power losses information and observe the deviation between the normal and abnormal state using neural networks techniques. The proposed monitoring and detecting method used real world data and also simulated data. This method was able to detect the abnormal behavior. Mehrotra et al. [8] proposed a method to classify the applications running in the smartphone using the power consumption data. This method classified the applications into three categories: low, medium and high. Their dataset was collected by installing an application in the smartphone to collect the consumption data. Then all data were divided into three types based on the power consumption. Classification was conducted by using five different techniques. Their method was applied on 77 applications and managed to successfully classify them as low, medium and high based on their power consumption. Kurniawan et al. [9] developed a detection method for Android smartphone operating system. The method was based on generating a dataset with sample period of one minute from the power consumption, the temperature of the battery and network traffic of the mobile. An application was used to log all the power data related to the battery. The researchers then classified the data, using four different machine learning algorithms. Their method achieved an accuracy of 85.6% with the Support Vector Machine (SVM) algorithm. According to Zeffere et al. [10], no access is needed to identify and detect the malwares. In their work, they proposed a methodology that detects the abnormal activities by analyzing the energy usage of an Android smartphone. Their dataset was generated by using a software which collects the consumption data with a sample period of 250 ms. Two techniques were used to classify the data and detect the malwares. This method detected malwares with 87% accuracy. Abykoon et al. [11] suggested a method which can identify what is the device connected to the power using the data of power consumption and machine learning techniques. Their dataset was collected with a sample period of five minutes. Classification in this research was using different supervised and unsupervised techniques. Each device had a unique signature that can be used to classify what type of device it is. However, in some cases and due to the similarities, more training was required in order to achieve high accuracy. 

Prediction of the needed power is important to ensure the stability of the power. Moon et al. [12] proposed a method that predicts the needed power using machine learning techniques. Their data were collected for one year with a 15-min sample period. In their model, they used two machine learning algorithms: Artificial Neural Networks and Support Vector Regression. The results of their approach show that Artificial Neural Networks had better accuracy than the Support Vector Regression. In many cases, the usage of the power differs between home use and industrial use, especially in terms of the rates and prices. Organizations such as electricity companies are trying to identify the behavior of the used power in order to ensure stable network and also differentiate between residential and commercial usage. Shao et al. [13] developed an energy consumption prediction model using Support Vector Machines to provide an estimation of the power consumption. Their dataset was collected by installing four sensors to measure the power, temperature, water flow and humidity. High accuracy prediction of the power consumption was successfully achieved and it included predictions such as possible high-power consumption during holidays at hotels and resorts. A method proposed by Gajowniczek et al. [14] attempted to identify if it is a household usage or not. Their dataset was provided by an electricity company for more than four thousand households with a thirty-minute sample period. They used different machine learning algorithms but they concluded that Support Vector Machine produced the best results with higher accuracy compared to the other algorithms. Markovič et al. [15], from the strategies of designing the electricity network, used classification techniques to predict the amount of power that will be needed in the future in order to avoid rebuilding the infrastructure. They collected a dataset of one year of power usage and they created different user profiles to classify the household users. Iqbal et al. [16] proposed an IoT based architecture capable of exploring electrical devices in smart homes, using sensors, load balancing and data processing. This architecture was tested on six hours of dataset and was proven successful. Data-driven models were used by Bourdeau et al. [17] to predict power consumption with different machine learning techniques and also used different techniques for data preprocessing. This is because it is mainly affecting the accuracy of the prediction results. Input data origin and variables of the dataset also affected the accuracy of the prediction results. The data were analyzed and divided into eight different types: building characteristics and operation information, deep-learning-based time series, mathematical characteristics, past time-steps/data points, time-related indicators, occupancy, indoor environment, and outdoor environment. The researchers concluded that different machine learning techniques were different for each scenario, with each of them having its own advantages and disadvantages.

Croce et al. [18] developed a new technique to monitor power consumption for smart buildings in which user privacy is respected. The architecture was designed as a distributed peer-to-peer to monitor power consumption, and connected with the nearest node, giving multiple peer-to-peer connections. Instead of using individual data, they clustered them to preserve privacy. However, the method had other features such as load control, energy prediction, and voltage stabilization. In this research, ready residential building datasets were used, and the sample period was of one week.

A new method was proposed by Zhang et al. [19] to monitor power consumption and then to store and analyze these data using machine learning. By utilizing the central processing unit (CPU), the consumption changed. Validation was completed by testing and simulating an attack that was intended to make a change on the CPU in order to create and simulate an utilization, and then the method could efficiently detect the change of power consumption with high accuracy and privacy. The authors used different simulation techniques and then analyzed and verified that the method is achieving high accuracy.

Youngjun and Young [20] proposed a new sensor that can measure high surges. Protection from surges is an advantage of this sensor in which it will represent the duration and lifetime of the protection with the help of an application that is used along with the sensor. The main idea of this research is to ensure that power is stable without any extra surge that affects the electrical device and will be damaged if it exceeds the normal limits. In this research, the sensor was designed as hardware with a small liquid-crystal display (LCD) screen in a small plastic enclosure box. The experiment was implemented using a surge simulator to increase the current for testing the sensor.

Blazakis et al. [21] discussed a new detection method for analyzing the power consumption data and then identify if there was any user that was not authorized and returned to fraud activity. An existing dataset was used in this research, which was generated from a real usage of more than five thousand houses for the duration of almost two years and the sample period for the dataset is half an hour. The results showed that this method has a successful detection rate compared to other methods, and almost in every scenario the method detected the unauthorized usage of the electricity. More than ten scenarios were tested to verify the success of this method.

Analysis and comparison of power consumption in different buildings were proposed by Cibinskiene et al. [22]. The importance of the energy savings and how to reduce energy consumption, especially in workplaces concerning previous researches to ensure continuity of the energy were studied. The regression analysis proved that saving energy will make a difference if the behavior of the staff focused on saving energy goals, and behavior was the most effective part for changing power consumption at residential or workplaces.

Jooseok Oh [23] discussed the use of smart technologies under the Internet of things that can save energy. Once home users can monitor and see the consumption of each device in their home, they will get more knowledge about the consumption of different devices, and this knowledge helps to shut down the devices that consume a lot of power and are not used, such as heating devices and air conditioners. A study was done by Jooseok to give Internet of things devices and training to home users to see if they can reduce usage and achieve power consumption. Home users successfully reduced the power consumption with the help of the Internet of things and the smart plugs, and this confirms that if users can see the real monitoring of the home appliances, they will manage their usage efficiently by using a timer and scheduling the time to start and stop.

Threats of Internet of things (IoT) devices have been highlighted by Myridakis et al. [24]. In this research, the dataset from the IoT devices was used with a detection system along with the difference in power current from the power supply to identify if there was an attack. One of the advantages is that this technique has the lowest costs. The experiment included a device that is simulated to be attacked by changing the hardware as an attacker change. The results of the experiment concluded with successful identification of security attack for the smart IoT devices as it is designed with basic features that may have an attack due to the simplicity.

Moradzadeh et al. [25] proposed a technique to monitor and analyze the power consumption data for the disaggregation and for calculating the expected power needed for the future for specific devices. The dataset was drawn from real data for power consumption captured for home electrical usage. The sample of the dataset was two days. Using the machine learning technique dataset was analyzed and validated the concept of the power consumption data, we can estimate the total needed power for each electrical device.

Forecasting electricity consumption using autoregressive integrated moving average (ARIMA) among industries at Guangdong province in China was performed [26] Three ARIMA models were considered to forecast electricity consumption. The research showed that ARIMA (1,1,1) provided results which are precise and could predict effectively. Similarly, variation in energy consumption for South Africa was predicted using ARIMA, Nonlinear Grey Model (NGM) and NGM-ARIMA models [27].

In order to achieve our objectives, we collected real power consumption data from a computer cluster of two machines as well as the consumption data from each machine, of a university computer lab for 48 h, using power monitoring sensors. In parallel, we obtained data from these machines regarding what applications are running during the experiment. During the last 24 h, we infected one of the machines with a custom virus (described in Section 2) in order to observe the variations in the power consumption. Upon completion, the three datasets were merged into one and processed using different machine learning techniques for the purpose of identifying associations between what was happening in the machines/cluster and the power consumption.

Making predictions based on power consumption profiles for applications is challenging due to the nondeterministic nature of the problem. A machine is running multiple applications and each application can consume different amount of power at different times based on activities. Attempting to speculate what is happening in a cluster of multiple computers adds additional complexity to the problem.

The rest of the paper is structured in the following way: Section 2 explains the experiment and data collection process in detail. Section 3 analyzes the data using various approaches. Section 4 concludes the work while providing directions for future research in this area.

## 2. Materials and Methods

This section explains the materials and methods relevant to each of the three proposed research objectives.

### 2.1. Experimentation Setup

In order to answer the research questions presented in the introduction, we generated our own dataset, which included various system characteristics that were collected while using the system with and without viral infection. The dataset associated what application was running and its corresponding power consumption. Both software and hardware solutions were utilized for this purpose.

The experiment utilized a computer cluster consisting of two Dell 9020 Desktop machines with dedicated NVIDIA GeForce 210 GPU cards. The machines were connected to three power consumption measurement sensors. Each machine and the cluster were connected to one SONOFF POW R2 sensors with the ESPurna firmware [28]. All machines and sensors were identical. A Raspberry Pi3 with NodeRed [29] acted as a MQTT broker server that will collect real-time data every second from the SONOFF sensors. The Sensors were connected to the Pi3 via Wi-Fi. Figure 1 illustrates the topology schematic of the experiment used for data collection and Figure 2 shows the connectivity between the devices.

The various devices participating in the experiment as well as their connectivity is illustrated in Figure 2. Table 1 presents the schedule of the applications that were executed during the experiment. As it can been seen from Table 1, various applications were run for different times segregated from each other. The different run durations (e.g., an application running for one hour and another one for 30 min) increases the difficulty of the analysis but it represents a more realistic usage.

During the second day at 11:30 AM, we infected desktop number 2 with a virus generated by the Virus Maker tool [30] and continued running the applications as normal. The Virus Maker tool has the ability to add multiple behaviors to infected files. However, in this experiment we were mainly interested in behaviors that potentially have impact to the power consumption. Such behaviors include, “random activity”, “infinity message boxes”, “opening of random files” and “slowing down of the computer”. Figure 3 shows the options selected during the generation of the virus.

During the experiment, three different datasets were generated: one from the sensor of the cluster and one from the sensor of each desktop. The files were merged to one unified dataset that was used in this research. The dataset (see Appendix A) contains eleven fields. The first field indicates the date and time of the recording. The second and fifth represent the IDs of the running applications of desktop 1 and 2. Fields three and seven contain the current (I) measured (in amperes) of the two desktop computers every second respectively. Fields four and eight contains the value of the power consumption (in Watts) of the two of the desktops every second. The sixth field, which is of Boolean type, represents the presence (1) or absence (2) of a virus. The current and power consumption of the cluster of the two computers every second, is shown in fields nine and ten. The dataset as described here shows that there are various parameters, like current and power, obtained from the cluster. These indicators are obtained from various nodes of the cluster in order to precisely identified the application and the presence or absence of the virus.

### 2.2. Proposed Methodology

The steps involved in the proposed methodology from preprocessing stage to the implications of various factors, along with time-series is described in this section. In order to achieve the objective 1 (“detect if a node and/or a cluster is infected”), descriptive analysis in the form of mean and standard deviation were used. To further emphasis our findings, variances of these measures were compared using an F-test. Objective 2 (“differentiate between different types of applications”), which is related to different applications, was analyzed using two-way ANOVA to understand whether there is an interaction between the virus and variety of applications. Furthermore, objective 3 (“detect occupancy of a node and/or a cluster), which is related to the occupancy of the node/cluster, can be predicted or identified using time-series based ARIMA method.

#### 2.2.1. Preprocessing

The obtained dataset is obtained from sensors, where outlines or invalid data is possible. The preprocessing of the dataset was performed manually by removing certain outlines or mistakes introduced during the data collection process, such as nonarrival of data from a sensor, negative values, etc.

#### 2.2.2. Descriptive Analysis

Various salient features such as mean and standard deviation were used to compare the factors involved in our experimentation. In order to validate the variance in the given factor, F-test and ANOVA are performed as described in Section 2.2.3 and Section 2.2.4

#### 2.2.3. Comparing Power Consumption with and without Virus

We studied the difference in variances between the power values when the virus is present or absent. Such an analysis is critical to recognize whether the interval (upper and lower bound) of power consumption for any specific application overlaps during the presence or absence of virus. Nonoverlapping of intervals with appropriate significance level will provide the necessary confidence to recognize the presence or absence of the virus. This was analyzed using a two samples of variances F-test with standard 95% significance level.

#### 2.2.4. Implications of Three Factors

Virus, application and power: having studied the differences in power consumption and the respective variances between different applications, with and without the running of the virus, we wished to see the implications of the factors involved in this study. Two-way ANOVA was used to analyze the dataset. As we have two factors (virus and application ID) involved, and the dependent variable is power. Data obtained from various sensors are analyzed further to validate the following hypotheses.
Mean values of the observations due to one factor (presence or absence of virus) are the same.Mean values of the observations due to another factor (difference in applications: multimedia related, office related, idle) are the same.There is no interaction between the two factors (presence/absence of virus and variety of applications).

#### 2.2.5. Time Series

In order to perform ARIMA [31], the following three parameters need to be obtained:p: The number of previous/lagged Y values considered in our model for each time point. p indicates autoregressive component;d: The number of differences considered in our model to follow stationarity;q: The number of previous/lagged error values considered in our model for each time point.

Autocorrelation functions (both complete and partial), named as ACF and PACF, were used to obtain these parameters. Meanwhile, automatic forecasting algorithms could be used to identify the parameters for ARIMA. Akaike’s information criterion (AIC) was used to select the appropriate model.
(1)$$AIC=L^{*}\left(\hat{\theta },{\hat{x}}_{0}\right)+2q$$
where *q* represents the number of parameters in *θ* plus the number of free states in $x_{0},\;\hat{\theta }\;$ and ${\hat{x}}_{0}$ are the estimates of *θ* and ${\hat{x}}_{0}$.

The choice of AIC, shown in Equation (4), over other measures like MSE (mean square error) or MAPE (mean absolute percentage error) was justified because AIC considers the likelihood instead of one-step forecasts. Also, AIC was suitable for the selection among additive and multiplicative models.

Nonseasonal ARIMA(p,d,q) is given as:(2)$$\phi \left(B\right)\left(1-B^{d}\right)y_{t}=c+\theta \left(B\right)\varepsilon_{t}$$
where {$\varepsilon_{t}$} is white noise, *B* is backshift operator, *ϕ(B)* and *θ(B)* are polynomials of order *p* and *q* respectively.

Seasonal ARIMA (p,d,q)(P,D,Q)_m_ is given as:(3)$$\Phi \left(B^{m}\right)\phi \left(B\right){\left(1-B^{m}\right)}^{D}{\left(1-B\right)}^{d}y_{t}=c+\Theta \left(B^{m}\right)\theta \left(B\right)\varepsilon_{t}$$
where *m* is the seasonal frequency, $\Phi \left(B^{m}\right)$ and $\Theta \left(B^{m}\right)$ are polynomials of order *P* and *Q* respectively.
(4)AIC=−2log(L)+2(p+q+P+Q+k)
where *L* is the maximum likelihood of the model fitted to the differenced data ${\left(1-B^{m}\right)}^{D}{\left(1-B\right)}^{d}y_{t}.$

## 3. Results and Discussion

### 3.1. Preprocessing

Before proceeding towards analyzing the obtained dataset, certain preliminary preprocessing revealed that, out of the 16 applications, along with an idle condition, all except one application was tested with both the presence and absence of virus. The application which represents opening the virus file could not be performed in the absence of the virus. The summary of the dataset collected is provided in Table 2. As an example, the number of records obtained from inactive virus scenario with an idle condition (application ID of zero) was 55630, while that of active virus scenario was 47929. In order to make the comparison effective, randomized selection of records for each application was performed. All the records were made equal to the record size of 494, which was the minimum record size available in the original dataset. The application number was 16, which means opening of the virus file was removed during the preprocessing stage.

### 3.2. Descriptive Analysis

As shown in Table 3, descriptive analysis of the obtained data was performed to find the average and standard deviation of current and power values obtained from all the three sensors. In order to make sure that there was no other factor involved in this analysis, data analyzed here were taken only from a computer. The objective of this task was to compare the average and standard deviation between two computers running the same application(s) but with and without the virus running on them. Outcome analysis revealed that the average and standard deviation of current and power differed when the virus was running. For example, when the virus was active, application ID 2 consumed an average power of 61.78 W while that of the inactive virus consumed 43.69 W. The range due to average +/- standard deviation are quite apart each other. This basically confirms that there was a clear difference in power consumption when virus was active or inactive in a system. Also, this could help in recognizing whether a computer is currently being used or not. In terms of power consumption, the applications consume higher to lower power in this order: multimedia and design applications, Microsoft Office applications and an idle condition.

### 3.3. Comparing Power Consumption with and without Virus

A sample of variance comparison for idle (no application) and application 15 is provided in Table 4. The comparison of F-critical with F (F ratio) value and the *p*-value with alpha, rejects the null hypothesis that their variances are the same. For example, in the case of idle condition, when no application was running, the F-critical value was 1.0123 and the F (F ratio) value was 2.6722. As the F-critical value is smaller than F (ratio) value, we conclude that the variances between the power with and without virus running is not the same. Also, *p*-value is smaller than the default alpha value of 0.05. Similar results are obtained from other applications as well.

### 3.4. Implications of Three Factors

Table 5 reveals the outcome of two-way ANOVA analysis. For the presence or absence of virus, when we compare F-Critical with F (F ratio), F-Critical (3.84) is smaller compared to F (F ratio) (2603.69), thus we reject the null hypothesis that the mean of the observations due to presence or absence of virus is not the same. Also, the obtained *p*-value is much smaller than that of alpha (0.05). This confirms our idea that the power consumption is affected due to running of virus program in a computer.

With regards to the variety among the applications, the F-critical (3.84) is smaller than the F (F ratio) value (302277.3). Thus, we reject the null hypothesis that the mean values of the applications are the same. It is clear that, from our study, the application is running at any point of time could easily be identified.

Similarly, the third hypothesis, which indicates that there is no interaction between virus and applications, is also rejected. Here, there is a clear indication that the impact of virus depends on the respective application. Thus, the rate of change in power consumption varies from one application to other, when the virus is executed.

In summary, the following is determined:Viruses have an impact on the power;A variety of applications impact the power consumption of the system;There is an interaction between the virus and the application with regards to the impact on the system.

A similar two-way ANOVA was performed after grouping the applications into nine groups like idle, programming, multimedia, design, general, etc. The outcome of such an analysis is provided in Table 6. Similar to the outcome in Table 5, Table 6 also rejects the null hypothesis. This confirms the impact of virus on power along with the respective group of application being executed.

### 3.5. Time Series

Having collected the data for two days while applying various activities with and without the virus, a time-series-based analysis was also performed. Time series is composed of trend, seasonal and cyclic components. Before performing ARIMA (autoregressive integrated moving average) on the time series, the assumption of stationarity was evaluated. An augmented Dickey–Fuller test was performed to test for stationarity. The results reveal the dataset, power consumption data specifically, follows stationarity, with a *p*-value of 0.01, which is less than the alpha value of 0.05. Smooth moving average was performed over the given power data. Averaging over a time period of 20 min returns the trend of the curve as in Figure 4.

The upward trend at a certain time indicates the impact of running a virus on the power consumption. Decomposition of the time-series into trend, seasonal, and random components reveals that there is generally a trend component in the obtained time-series data. Excluding trend, the seasonality is not prevalent in the dataset. Also, the random component does not seem to highly impact on the data. Figure 5 shows the decomposition of additive time series.

Based on our experimental analysis, ARIMA (2,0,0) (1,1,0) [60] with drift was found to be the most suitable model. This model obtained an AIC of 792012.8. The value of *p* was 2, which indicates the relationship between an observation with that of the one preceding the one immediately preceding it. The zero value of d indicates that the integrative component was absent. The value of zero for q indicates no relationship between an error and the ones preceding it. The presence of seasonality in the model indicates repetitions of the same events. In our analysis, the repetitions or seasonality depends on the time at which certain applications or the virus were executed. As in real time such an event varies from one situation to another, the seasonality value indicated by the proposed model was not significant in this study.

Forecasting for a time period of 3000 s was done using the selected model as shown in Figure 6. Plot shows that the pattern is expected to remain in the same level with the upper and lower bound tending to vary.

### 3.6. Outcomes from the Analysis

Overall, this study has provided the following insights:Descriptive analysis: Average and standard deviation of power consumption varied while running any specific application, with or without virus;F-Test of two samples of variances: variances of power consumption varied with the presence or absence of virus;Two-way ANOVA: The interaction between the presence or absence of virus and the specific application running does impact the power consumption on the computer. Power consumption of the computer also varied due to the type of the application and whether the virus was running or not;Time-Series: Time series analysis on the dataset reveals that the power consumption can be represented with ARIMA model using autoregression.

## 4. Conclusions

This paper presented a methodology for detecting potentially infected computers and cluster of computers connected on a smart grid by analyzing their power consumption data. The research used data of a cluster of two computers connected to power measurements sensors. The data were analyzed using descriptive analysis, two-way ANOVA and ARIMA. The results showed that the presence/absence of virus and its interaction with variety of applications does affect the power consumption. We were able to detect infected nodes and clusters. Additionally, we can detect what type of application is running on the machine. It is possible to accurately detect physical or remote occupancy of the lab as there is a significant variation of the power usage. All these points make this work useful for facility management, lab administrations and many more. This research can be generalized to any context with the proposed experiment architecture capable of facilitating individualistic data acquisition, hence allowing the system to produce diagnostics. The main limitation of this work lies with the number of executed applications during the experiment and the limited number of nodes and duration. However, the aim of this research was to test the feasibility of such an approach, something that was proven based on the above results. Future work will mainly focus on enhancing the dataset to overcome the above limitations. Additionally, we will detect normal and abnormal behaviors, for example, massive sudden power consumption from a PC that is currently used is a suspicious behavior.

## Figures and Tables

**Figure 1 sensors-20-05075-f001:**
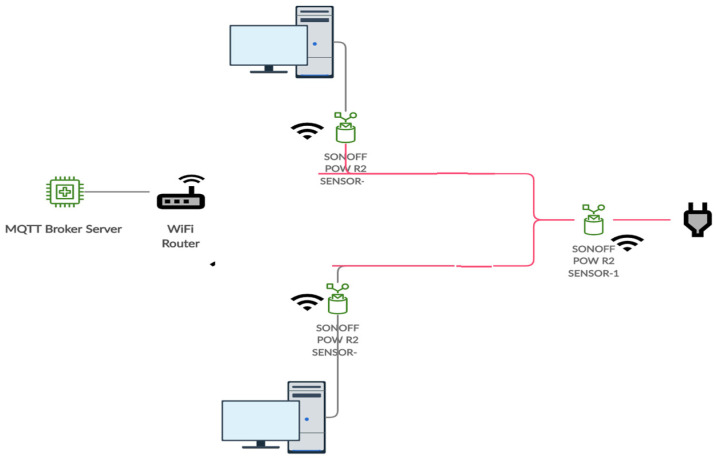
Experiment topology schematic diagram.

**Figure 2 sensors-20-05075-f002:**
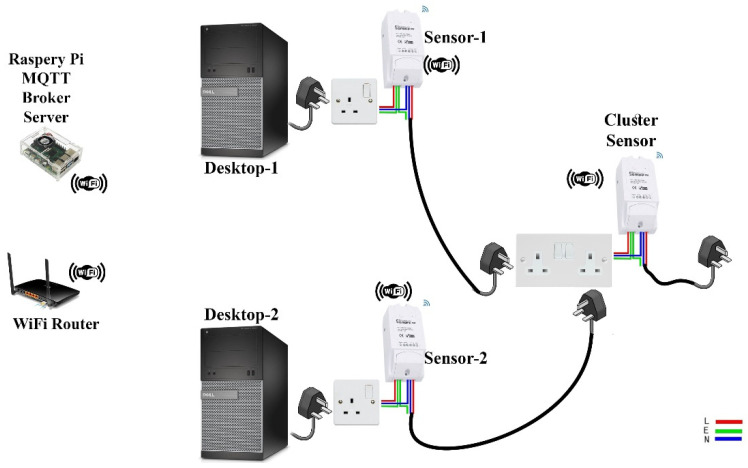
Experiment device connectivity.

**Figure 3 sensors-20-05075-f003:**
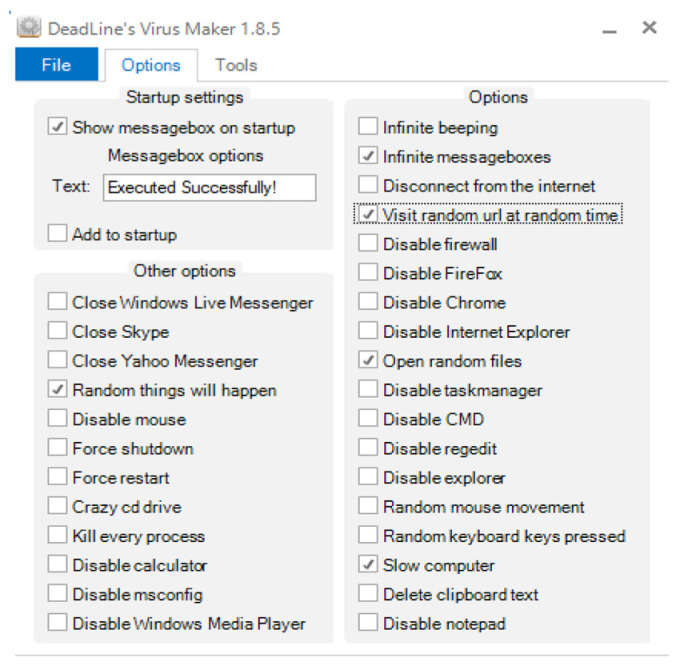
Screenshot of DeadLine Virus Maker tool.

**Figure 4 sensors-20-05075-f004:**
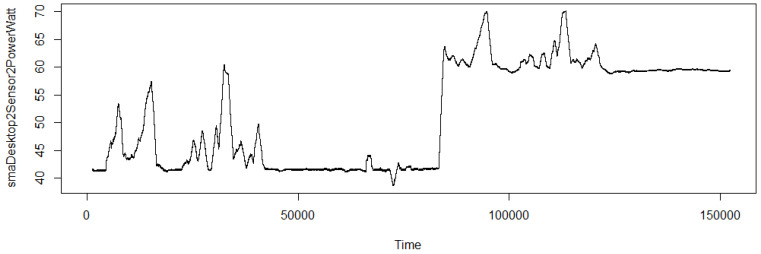
Smooth moving average with n = 1200 (20 min).

**Figure 5 sensors-20-05075-f005:**
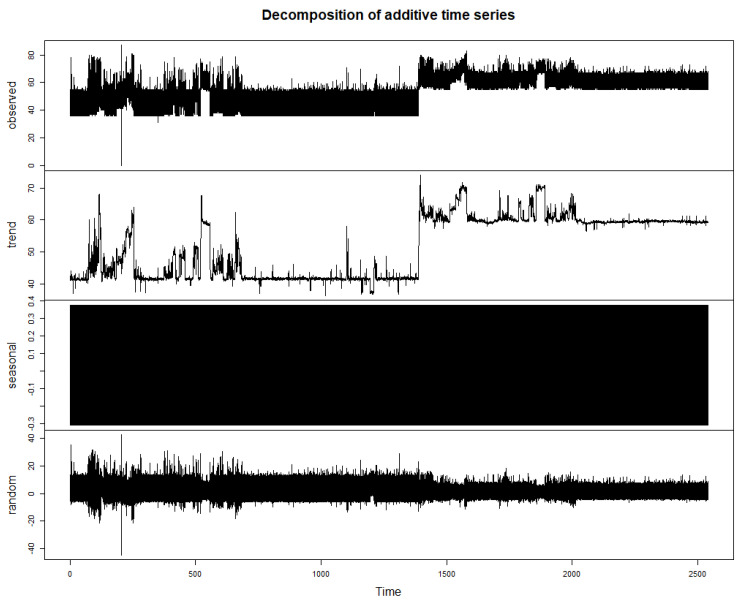
Decomposition of the time series dataset.

**Figure 6 sensors-20-05075-f006:**
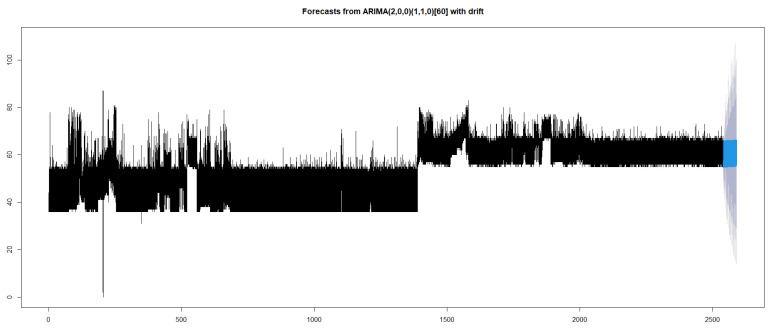
Forecast using ARIMA.

**Table 1 sensors-20-05075-t001:** Application execution schedule.

#	Application Name/Action/Process	1st Day		2nd Day	
		Start Time	End Time	Start Time	End Time
1	Start of Experiment	05:01		05:35	
2	Open the virus on desktop 2–Sensor 2	-	-	08:00	
3	NetBeans	08:00	09:00	08:05	09:00
4	Idle	09:00	09:10	09:00	9:10
5	MS Word	9:10	10:00	9:10	10:00
6	Idle	10:00	10:07	10:00	10:07
7	YouTube 240p	10:07	10:20	10:07	10:20
8	YouTube 360p	10:21	10:35	10:21	10:35
9	YouTube 1080p	10:36	10:50	10:36	10:50
10	YouTube 1440p	10:51	11:15	10:51	11:15
11	YouTube 4K	11:15	11:25	11:15	11:25
12	Idle	11:25	11:30	11:25	11:30
13	MS Excel	11:30	12:00	11:30	12:00
14	Idle	12:00	13:35	12:00	13:35
15	3D Builder (Paint 3D)	13:35	14:20	13:35	14:20
16	Idle	14:21	14:40	14:21	14:40
17	Copy Large Files 1st Half From USB to PC	14:40	15:05	14:40	15:05
18	Idle	15:05	15:40	15:05	15:40
19	Copy Large Files 1st Half From PC to USB	15:40	16:00	15:40	16:00
20	Idle	16:01	16:10	16:01	16:10
21	Game (The great unknown Houdini’s castle) (*Game Crashed at Desktop 2–Then Opened again with no issue*)	16:11	16:50	16:11	16:50
22	Idle	16:51	17:00	16:51	17:00
23	Fusion 360	17:00	17:41	17:00	17:41
24	Idle	17:41	18:00	17:41	18:00
25	MS PowerPoint	18:00	18:30	18:00	18:30
26	Idle	18:31	18:35	18:31	18:35
27	Web Browsing	18:36	19:00	18:36	19:00
28	Idle	19:00	05:35	19:00	05:35
29	End of Experiment		05:35		05:35

**Table 2 sensors-20-05075-t002:** Summary of dataset.

Original Dataset			Preprocessed Dataset		
Application ID	Inactive Virus	Active Virus	Application ID	Inactive Virus	Active Virus
0	55,630	47,929	0	494	494
1	3123	3235	1	494	494
2	2624	2869	2	494	494
3	771	704	3	494	494
4	736	889	4	494	494
5	846	884	5	494	494
6	1216	1173	6	494	494
7	597	494	7	494	494
8	1626	1715	8	494	494
9	2576	2674	9	494	494
10	1160	1179	10	494	494
11	1410	1416	11	494	494
12	2338	2029	12	494	494
13	2387	1934	13	494	494
14	1524	1823	14	494	494
15	1475	1282	15	494	494
16	0	126			

**Table 3 sensors-20-05075-t003:** Descriptive analysis.

Application ID	AVE (No Virus): Current	STDEV (No Virus): Current	AVE (Virus): Current	STDEV (Virus): Current	AVE (No Virus): Power	STDEV (No Virus): Power	AVE (Virus): Power	STDEV (Virus): Power
0	0.274250333	0.028103	0.347148	0.020521	41.65125	7.485862	60.35578	4.995615
1	0.30206212	0.040198	0.279725	0.032956	49.17003	10.83542	43.31901	8.913672
2	0.282293826	0.029435	0.351423	0.019544	43.69131	7.785045	61.7839	5.003486
3	0.294744488	0.027763	0.349382	0.016515	47.10895	7.176698	60.8821	4.130037
4	0.338485054	0.282622	0.352319	0.018267	47.13043	9.747805	61.50731	4.612483
5	0.310338061	0.027774	0.347696	0.017278	51.02364	7.14727	60.18891	4.235889
6	0.328525493	0.02398	0.351409	0.018236	55.54605	6.243444	60.99829	4.507483
7	0.336469012	0.045425	0.360682	0.014323	59.01173	12.08047	63.31984	3.612988
8	0.276551661	0.028898	0.374414	0.015795	42.26999	7.718847	66.68105	3.88115
9	0.285190606	0.031642	0.346214	0.017018	44.44293	8.223566	59.27786	4.086996
10	0.304116379	0.030956	0.355818	0.020229	49.19138	7.936956	61.79389	4.938064
11	0.299434043	0.029658	0.352472	0.019252	48.02128	7.655155	60.73729	4.650185
12	0.342401625	0.025261	0.353526	0.020993	58.9337	6.272336	61.61656	5.078656
13	0.289771261	0.030464	0.374752	0.023172	45.81232	8.066154	66.72285	5.671899
14	0.282875328	0.029596	0.350807	0.018327	43.95276	8.015767	61.19748	4.584067
15	0.300162712	0.035122	0.345573	0.017133	48.56271	9.288857	59.97738	4.294064

**Table 4 sensors-20-05075-t004:** F-Test of two samples of variance.

Variance of Power in Idle Condition			Variance of Power with Application 3		
	Variable 1	Variable 2		Variable 1	Variable 2
Mean	39.93072	60.02728	Mean	58.67635	60.01577
Variance	112.9857	42.28129	Variance	42.94666	18.63846
Observations	72,354	72,354	Observations	1205	1205
df (Degrees of Freedom)	72,353	72,353	df (Degrees of freedom)	1204	1204
F (F ratio)	2.672238		F (F ratio)	2.304196	
P(F< = f) one-tail	0		P(F< = f) one-tail	1.53 × 10^−46^	
F-Critical one-tail	1.012305		F-Critical one-tail	1.099489	

**Table 5 sensors-20-05075-t005:** Two-way ANOVA.

ANOVA						
Source of Variation	SS (Sum of Squares)	df (Degrees of Freedom)	MS (Mean Squares)	F (F Ratio)	*p*-Value	F-Critical
Virus	150,880.9	1	150,880.9	2603.692	0	3.841753
Application	17,516,617	1	17,516,617	302277.3	0	3.841753
Interaction	104,200.9	1	104,200.9	1798.154	0	3.841753
Within	1,831,878	31,612	57.94883			
Total	19,603,577	31,615				

**Table 6 sensors-20-05075-t006:** Two-way ANOVA.

ANOVA
Source of Variation	SS (Sum of Squares)	df (Degrees of Freedom)	MS (Mean Squares)	F (F Ratio)	*p*-Value	F-Critical
Virus	2429.457	1	2429.457	39.57663	3.21 × 10^−10^	3.841853
Application	17,168,455	1	17,168,455	279679.6	0	3.841853
Interaction	2429.457	1	2429.457	39.57663	3.21 × 10^−10^	3.841853
Within	1,448,467	23,596	61.38615			
Total	18,621,781	23,599

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
