# Peer review of "Detection of Potentially Compromised Computer Nodes and Clusters Connected on a Smart Grid, Using Power Consumption Data"

_sensors, 2020, doi:10.3390/s20185075_

Round 1

Reviewer 1 Report

The topic is interesting and it is adapt to this journal. The collaboration among several faculties is useful and I think that there is a great work behind the presentation of this work. However, while the presentation is nice in shape, there are few comments and/or suggestions to improve the manuscript.

-According to scientific standards, abbreviations cannot be used in the abstract, please correct it in the manuscript.

-Clarify better the innovation of this work in the abstract and in the main text.

-Read articles to understand the structure of Sensors. The following structure would be preferable based on the Sensors Microsoft Word template file: 1. Introduction (1.1, 1.2, 1.3.), 2. Materials and Methods (2.1, 2.2., 2.3.), 3. Results and Discussion (3.1, 3.2, 3.3), 4. Conclusions. These sections mixed in the text. The introduction section is a literary review of the topic. In the introduction (or where still necessary) all paragraph must be cited because of the risk of plagiarism.

https://www.mdpi.com/journal/sensors/instructions

-A short paragraph introducing the problem statement and actions taken (or a description of the study) should be included at the end of the Introduction section.

-It would be important to compare the results with other „new” modeling concepts.

-Please add more references because the number of scientific references is low. Please provide more general information on the importance of this topic in the introduction. At least 25-30 scientific manuscript need to use because this is a Q1 Journal! What is the role of this topic in literature and in international context? 

-The citation style is inappropriate. Mendeley can easily solve this problem. In Mendeley can be found also the style of Energies (Mendeley is a free reference manager and an academic social network): https://www.mendeley.com/ 

-Please add more information's about the model validation in a new chapter. Please better support the accuracy of the model with measurement results.

The manuscript must be at least 15-20 pages long.

-Extend the conclusion with more general usability. What are the benefits of the results in a global context? Please explain this better in the manuscript.

Author Response

Reviewer 1

Comments and Suggestions for Authors

The topic is interesting and it is adapt to this journal. The collaboration among several faculties is useful and I think that there is a great work behind the presentation of this work. However, while the presentation is nice in shape, there are few comments and/or suggestions to improve the manuscript.

-According to scientific standards, abbreviations cannot be used in the abstract, please correct it in the manuscript.

Addressed

-Clarify better the innovation of this work in the abstract and in the main text.

The paper was restructured to make the novelty and contributions more clear. In more detail:

“The results show that this is possible to detect what type of application is running and if an individual machine or its cluster are infected. Additionally, we can conclude if the lab is used or not, making this research an ideal management tool for administrators.”

“The novelty of this work can be summarized by the following objectives:

  • Detect if a node and/or a cluster is infected
  • Differentiate between different types of applications
  • Detect occupancy of a node and/or a cluster”

-Read articles to understand the structure of Sensors. The following structure would be preferable based on the Sensors Microsoft Word template file: 1. Introduction (1.1, 1.2, 1.3.), 2. Materials and Methods (2.1, 2.2., 2.3.), 3. Results and Discussion (3.1, 3.2, 3.3), 4. Conclusions. These sections mixed in the text. The introduction section is a literary review of the topic. In the introduction (or where still necessary) all paragraph must be cited because of the risk of plagiarism.

https://www.mdpi.com/journal/sensors/instructions

 The paper is now restructured according to the template

-A short paragraph introducing the problem statement and actions taken (or a description of the study) should be included at the end of the Introduction section.

“The steps involved in the proposed methodology from pre-processing stage to the implications of various factors, along with time-series is described in this section. In order to achieve the objective 1 (“Detect if a node and/or a cluster is infected”), descriptive analysis in the form of mean and standard deviation were used. To further emphasis our findings, variances of these measures were compared using F-test. Objective 2 (Differentiate between different types of applications), which is related to different applications, was analyzed using Two-way ANOVA to understand whether there is an interaction between the virus and variety of applications. Furthermore, the Objective 3 (Detect occupancy of a node and/or a cluster), which is related to the occupancy of the node/cluster, can be predicted or identified using time-series based ARIMA method.”

-It would be important to compare the results with other „new” modeling concepts.

This paper utilized a dataset that was created for this research hence direct comparison is not possible. Also it is trying to solve a novel problem.

-Please add more references because the number of scientific references is low. Please provide more general information on the importance of this topic in the introduction. At least 25-30 scientific manuscript need to use because this is a Q1 Journal! What is the role of this topic in literature and in international context? 

Done

-The citation style is inappropriate. Mendeley can easily solve this problem. In Mendeley can be found also the style of Energies (Mendeley is a free reference manager and an academic social network): https://www.mendeley.com/ 

 Done

-Please add more information's about the model validation in a new chapter. Please better support the accuracy of the model with measurement results.

Done. The paper has been extended and it now includes all this information..

The manuscript must be at least 15-20 pages long.

done

-Extend the conclusion with more general usability. What are the benefits of the results in a global context? Please explain this better in the manuscript.

The conclusion has been modified.

Reviewer 2 Report

The motivation for this work and the background of available techniques are not clearly described in the introduction. In principle, the selected approach and theory are poorly described, though some improvements are necessary. The paper sections are less well written and need improvement, which might have an impact on the conclusions section. In specific, it remains unclear how to detect if a machine and a cluster of machines is infected by unwanted applications, such as a virus. In particular, the following points need to be improved:

  1. The motivation for this work and the background of available techniques is not clearly described in the introduction. In principle the selected approach must be revised.
  2. This reviewer cannot distinguish the new findings of this paper and the existing approaches in the literature. Instead, this reviewer suggests the authors to clearly mention what is new compared with existing approaches and why the proposed alternative is needed to be used instead of the existing methods. The author does not consider the existing studies from the literature, but the discussion is required along with the novelty of the proposed method. However, the novelty introduced is very poor and some important issues should be reconsidered.
  3. Please reconsider the comments for results and compare the proposed approach with others from the literature. In this way, the results have de guaranteed conclusions.
  4. Your proposed method considers the “outliers” from the power consumptions? Please explain the aforementioned concept.
  5. Also, in the paper title, the author considers the smart grids, but in the paper text this aspect is missing. Please explain why the two computers are considered as smart grids. I think that the title must be modified accordingly.
  6. There are many limitations of the proposed method. One of them are that this application only process the power consumption.
  7. In conclusions, the article has serious flaws and can be rejected, additional experiments must be concretely presented. The references are too few for a MDPI Journal. Also, please consider other relevant reference from the MDPI.
  8. This research is not correctly conducted. Please consider the first 5 suggestions!

The authors must strongly consider the aforementioned suggestions and resubmit.

Author Response

The motivation for this work and the background of available techniques are not clearly described in the introduction. In principle, the selected approach and theory are poorly described, though some improvements are necessary. The paper sections are less well written and need improvement, which might have an impact on the conclusions section. In specific, it remains unclear how to detect if a machine and a cluster of machines is infected by unwanted applications, such as a virus. In particular, the following points need to be improved:

  1. The motivation for this work and the background of available techniques is not clearly described in the introduction. In principle the selected approach must be revised.

The introduction has been modified to accommodate the request.

  1. This reviewer cannot distinguish the new findings of this paper and the existing approaches in the literature. Instead, this reviewer suggests the authors to clearly mention what is new compared with existing approaches and why the proposed alternative is needed to be used instead of the existing methods. The author does not consider the existing studies from the literature, but the discussion is required along with the novelty of the proposed method. However, the novelty introduced is very poor and some important issues should be reconsidered.

The introduction has been modified. Also the paper has been restructured.

  1. Please reconsider the comments for results and compare the proposed approach with others from the literature. In this way, the results have de guaranteed conclusions.

This paper utilized a dataset that was created for this research hence direct comparison is not possible. Also it is trying to solve a novel problem. However, we included the dataset produced as part of this work. Hence also our experiments and results are repeatable. Also F-test and Anova were used to validate the results. Both the production of a new dataset and the validation add to the scientific contribution of this work.

  1. Your proposed method considers the “outliers” from the power consumptions? Please explain the aforementioned concept.

Handled in the pre-processing stage. It is now more clear as the paper has been restructured.

  1. Also, in the paper title, the author considers the smart grids, but in the paper text this aspect is missing. Please explain why the two computers are considered as smart grids. I think that the title must be modified accordingly.

We are using smart grid devices (such as sonoff sensors) and by definition our network is a smart grid. It is a small topology as it aims to be used as a proof of concept. However, as we explain it can be generalized to any company smart grid (local smart grid only).

  1. There are many limitations of the proposed method. One of them are that this application only process the power consumption.

Yes. We only focus on power consumption as all computational operations consume power. There is a lot of research in that areas (see introduction). Additionally, this is very challenging as explained in section 1. Hence it increases the scientific value of this work.

  1. In conclusions, the article has serious flaws and can be rejected, additional experiments must be concretely presented. The references are too few for a MDPI Journal. Also, please consider other relevant reference from the MDPI.

The paper including the conclusions has been modified.

  1. This research is not correctly conducted. Please consider the first 5 suggestions!

Addressed

Reviewer 3 Report

Dear Authors

The information in this paper (Identification of potentially infected nodes and clusters connected on a smart grid, using power consumption data and machine learning) is not sufficient enough and it is advised that you’d hold the publication at this time.

However, the motivation and results are considered to be worthy of a Major revision under the following conditions.

*1. (Important) This paper is short for journal publication.
Write more...

2. (Important) Abstract and contribution should be supplemented, making them clearer.
-It is almost impossible to understand the contribution of the paper.

3. The introduction section requires rewriting or supplementation.

4. (Important) Improve the related work...
-References must be more the 20 papers published from 2017~2020 [ex 21-40...] by major publishers such as Springer, Elsevier, IEEE, ACM, MDPI and Wiley.

5. (Important) Figure 1: Experiment setup is Poor.
5.1. Identification of potentially infected nodes
5.2. Clusters connected

6. The ‘Conclusion and Future Work’ part is short so that more details are required.
1) Need Future Work
2) Contribution

Anyway, the strength of the paper included:
the topic is interesting.

Author Response

Dear Authors

The information in this paper (Identification of potentially infected nodes and clusters connected on a smart grid, using power consumption data and machine learning) is not sufficient enough and it is advised that you’d hold the publication at this time.

However, the motivation and results are considered to be worthy of a Major revision under the following conditions.

*1. (Important) This paper is short for journal publication.
Write more...

The paper has been restructured and extended.

  1. (Important) Abstract and contribution should be supplemented, making them clearer.
    -It is almost impossible to understand the contribution of the paper.

The paper has been restructured to increase readability. Also clear statements were added:

“The results show that this is possible to detect what type of application is running and if an individual machine or its cluster are infected. Additionally, we can conclude if the lab is used or not, making this research an ideal management tool for administrators.”

“The novelty of this work can be summarized by the following objectives:

  • Detect if a node and/or a cluster is infected
  • Differentiate between different types of applications
  • Detect occupancy of a node and/or a cluster”
  1. The introduction section requires rewriting or supplementation.

Addressed

  1. (Important) Improve the related work...
    -References must be more the 20 papers published from 2017~2020 [ex 21-40...] by major publishers such as Springer, Elsevier, IEEE, ACM, MDPI and Wiley.

Addressed

  1. (Important) Figure 1: Experiment setup is Poor.
    5.1. Identification of potentially infected nodes 
    5.2. Clusters connected

Addressed

  1. The ‘Conclusion and Future Work’ part is short so that more details are required.
    1) Need Future Work
    2) Contribution

Addressed

Round 2

Reviewer 2 Report

The authors made a significant effort to respond to my suggestions. So, the quality of the paper increased substantially, and in my opinion, the manuscript is ready for publication if some minor revisions will be solved.

  1. The equation and expressions from sections 2.2 must be numbered.
  2. The tables and the titles of the table must be revised in accordance with the Journal template.
  3. Moreover, some coefficients are not explained. Please see Tables 4, 5 and 6 (SS, MS, df etc.)
  4. Please use a single notation for F-Critical, because in the paper the authors use "F crit", "F critical", "F-Critical" or "F Critical".

Author Response

Reviewer 2

The equation and expressions from sections 2.2 must be numbered.

Addressed. All equations and expressions are now number. Also the numbering has been updated to reflect the changes.

The tables and the titles of the table must be revised in accordance with the Journal template.

Addressed. All tables have been restructured based on the journal template.

Moreover, some coefficients are not explained. Please see Tables 4, 5 and 6 (SS, MS, df etc.)

All coefficients are now explained. We enable changes tracking in the word document so the changes can spotted easily.

Please use a single notation for F-Critical, because in the paper the authors use "F crit", "F critical", "F-Critical" or "F Critical".

Done. We now use F-Critical in all instances.

Reviewer 3 Report

Dear Authors.

The revision adequately address the concerns expressed in last review.
So, I recommend that this revised manuscript can now be recommended for publication (Accept as is: Accept in present form).

Author Response

Reviewer 3

The revision adequately address the concerns expressed in last review. 
So, I recommend that this revised manuscript can now be recommended for publication (Accept as is: Accept in present form).

Thank you